# Fatigue Properties and Residual Stresses of Laser-Welded Heat-Resistant Pressure Vessel Steel, Verification on Vessel Model

Jiří Čapek [1,*], Jan Kec [2], Karel Trojan [1], Ivo Černý [2], Nikolaj Ganev [1], Kamil Kolařík [1] and Stanislav Němeček [3,4]

1   Department of Solid State Engineering, Faculty of Nuclear Sciences and Physical Engineering, Czech Technical University in Prague, Trojanova 13, 120 00 Prague, Czech Republic
2   Strength Laboratory, SVÚM a.s., Tovární 2053, 250 88 Celakovice, Czech Republic
3   RAPTECH s.r.o., U Vodárny 473, 330 08 Zruc-Senec, Czech Republic
4   Department of Material and Engineering Metallurgy, Faculty of Mechanical Engineering, University of West Bohemia, Univerzitní 22, 301 00 Pilsen, Czech Republic
*   Correspondence: jiri.capek@fjfi.cvut.cz; Tel.: +420-224-358-624

**Abstract:** Most power plants use the Rankine cycle, where the heat supplied to water and steam is converted into mechanical work; therefore, most components have to be made of heat-resistant steel. Sufficient mechanical properties must be ensured for welded pipes to meet stringent requirements. Therefore, laser-welded 5 mm thick heat-resistant pressure vessel steel plates were subjected to various mechanical tests, including high-cycle fatigue tests. The microstructural notches were determined using X-ray diffraction too to determine critical areas that are susceptible to crack initialization and affect the service life. Finally, a functional model of the pressure vessel subsequently verified the results and assumptions. The presented results ensure the transferability of the results to real-life applications and outline the promising application potential of laser welding for producing vessels and pipes from heat-resistant steel in the industry.

**Keywords:** laser welding; heat-resistant pressure vessel steel; microstructure; X-ray diffraction; high-cycle fatigue tests

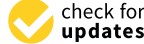



## 1. Introduction

Welding has been widely used in the fabrication industries, producing ships, trains, steel bridges, pressure vessels, and more, for decades. Increasing demands are being placed on the mechanical properties and durability of welds and, further, there is an increasing demand for productivity and cost effectiveness of welding, which leads to the use of modern and progressive methods, including laser welding. The power density of the laser beam is much higher than that of an arc, so laser welding shows higher welding speeds, productivity, deeper and narrower welds, and the possibility of joining components with a wide range of thicknesses (0.01 to 50 mm), mostly without additional material. The laser head is usually on the arm of the robot, which ensures precision and increases productivity even more [1,2].

Most power plants use the Rankine cycle, where the heat supplied to water and steam is converted into mechanical work; therefore, most components have to be made of heat-resistant steel. Ferritic heat-resistant pressure vessel steels are used, in particular, to manufacture boilers, pressure vessels, and pipes that transport hot liquids [3]. These steels are subject to high demands on creep resistance as a result of high temperature and internal overpressure. Because of the requirement for higher efficiency and service life of energy or chemical equipment, higher demands are also placed on the material from which the individual parts are made. Laser welding, with the advantages described above, could find an application in the welding of vessels and pipes instead of seamless pipes. Contrary to a seamless pipe manufacturing process, laser welding is considerably less

expensive. Generally, the welding joint is the weak point where quality affects overall performance. Nevertheless, the strength of the weld can be improved by choosing suitable welding parameters.

With the application of new welding technology, it is necessary to describe the influence on the basic mechanical parameters. Metallography, tensile, and hardness tests are usually an integral part of the set of performed tests. Furthermore, fatigue testing is essential to ensure safe and reliable service, mainly of dynamically loaded structures. Moreover, an investigation of fatigue properties is still missing in some works aimed at laser welding, e.g., high-pressure pipelines [4].

It is necessary to mention that the final microstructure and mechanical properties are strongly dependent on the welding parameters [5,6]. It was shown that laser power reduction decreases the residual stresses (RS) with hardness and increases the toughness. However, it is important to monitor sufficient penetration and pressure fluctuations. According to the previous work of Čapek et al. [7], the laser welding of low-alloy steel causes the generation of acicular ferrite in the weld metal (WM) and heat-affected zone (HAZ). The increase in hardness, the reduction in impact toughness, and the coarsening of the microstructure in comparison with the base metal were found. Moreover, using not only metallography but non-destructive XRD (RS, microstrain, and crystallite size), so-called notches were found. These microstructural and/or surface notches are the critical areas for potential crack initialization, e.g., at the interface among the WM and the HAZ in this case. Moreover, from the point of view of RS and fatigue, the advantage of laser welding compared to arc welding was definitely proven [7,8].

Although RS have not yet belonged to standard parameters for the certification of welding processes, they can be considered a significant characteristic of weld quality [9]. The aim of the laser welding process is not only to produce a joint but also to avoid the generation of undesirable stresses that would significantly promote crack propagation and cause distortion of the final object. Because of the heterogeneous application of energy and localized fusion, which occur during the welding process, a high undesirable RS can be present in the region near the weld and in the weld itself. These RS occur as a result of the superposition of thermal (tensile effect due to shrinkage) and transformation (compressive effect due to phase transformation from fcc lattice to bcc/bct lattice) stresses and could reach high values and, subsequently, could cause a reduction in fatigue life or promote brittle fractures [10,11]. Therefore, it is necessary to reduce the RS values and the size of the area affected by the welding. RS play one of the main crucial roles from a technological point of view, as they influence corrosion resistance, fatigue life, and other utility properties. Generally speaking, compressive RS decelerate crack nucleation, formation, propagation, and, for this reason, prolong fatigue life [11].

The stability of the welding process, the influence of welding techniques on RS, and the microstructural parameter FWHM (full width at half maximum) were investigated by Čapek et al. [8]. Laser welds have been shown to exhibit properties better than conventional manual arc welding (MAG) and comparable with electron welding. In the contribution of Černý et al. [12], the high-cycle fatigue resistance of the laser welds was detailed and discussed. The other part of the contribution was focused on the crack initiation mechanisms and welding imperfections, which are the most critical place for crack initialization, partly because of the reduction in the real effective cross-section and, therefore, the significant reduction in fatigue life.

In [13,14], the influence of a low- and/or high-cycle fatigue test on microstrain or the FWHM parameter was investigated. Cyclic hardening is exhibited by increasing FWHM and microstrain. Decreases in FWHM and microstrain are caused by cyclic softening explained by the movement of dislocations, their recombination, and the formation of different dislocation microstructure layout, which is a very long-term process. Therefore, the major decrease/relaxation in microstrains is caused by the total specimen disruption as a result of the long main fatigue crack and final failure. The main conclusion was that

the microstrain could be a good indicator of non-destructive microstructural changes, i.e., future crack initialization and propagation.

It is generally known that the integrity of welded structures is affected by two fundamental factors—the quality and geometry of the welds. If the welded structures contain harmful defects, such as lack of fusion or penetration, the service life can be significantly reduced. One of the most important geometric welding parameters is the undercut of the weld toe, which causes a reduction in fatigue life. In welds, axial misalignment occurs due to the disjointed position of the two parts to be welded or angular misalignment due to high post-weld tensile stresses induced by bending stresses. These and other types of influences also affect laser welding; therefore, this investigation aimed to contribute to the problems of integrity of the welded structures under fatigue loading, with particular attention to the effects of laser welding of pressure vessel steels and connected problems of RS occurring near the laser welds. Our study provides a complex, deeper, and comprehensive evaluation of the mechanical and fatigue properties of laser-welded pressure vessel steel plates. Specifically, 5 mm thick square butt welds made of P265GH steel plates for high-pressure vessels and boilers were analyzed. This fine-grained low-alloy carbon steel is widely used for the fabrication of high-pressure vessels, steam boiler parts, pressure piping, compressors, etc. The results and assumptions were subsequently verified on a functional model of the pressure vessel, which was analyzed and subsequently subjected to cyclic fatigue loading by internal overpressure with burst at the end.

## 2. Materials and Methods

First, P265GH heat-resistant pressure vessel steel plates were studied. Flat plates were used to better process the results, especially fatigue tests. The chemical composition of this steel is given in the European Standard EN 10028-2:2017 [15]; see Table 1. According to the expected maximum internal pressure during the pressure fatigue test of the model vessel (16 bar), 5 mm thick steel plate was selected.

**Table 1.** Chemical composition of the P265GH steel according European Standard EN 10028-2:2017.

| Element | Fe | C | Mn | Si | P | S | N | Al | Cu, Cr, Ni | V | Nb, Ti | Mo |
|---|---|---|---|---|---|---|---|---|---|---|---|---|
| Weight fraction (wt.%) | balanced | max. 0.2 | 0.8–1.4 | max. 0.4 | max. 0.025 | max. 0.01 | max. 0.012 | min. 0.02 | max. 0.3 | max. 0.02 | max. 0.03 | max. 0.08 |

The square butt weld (without wire) connecting two plates of dimensions 150 mm × 300 mm × 5 mm, see Figure 1a, was performed by co-author S. Němeček at RAPTECH s.r.o. with Laserline LDM3000-60 (Laserline GmbH, Koblenz, Germany). The most important welding parameters, such as laser power $P$, welding speed $v$, welding mode, laser wavelength $\lambda$, beam quality, beam spot, focusing distance, and shielding gas with its flow are given in Table 2 where the laser beam orientation was perpendicular to the plate. Negative defocus was used to achieve the maximum penetration depth; the focal point was 1 mm below the upper surface of the welded plate.

**Table 2.** Welding parameters.

| $P$ (W) | $v$ (mm·s$^{-1}$) | Mode | $\lambda$ (nm) | Beam Quality (mm·rad) | Beam Spot (mm) | Focusing Distance (mm) | Shielding Gas; Flow (L·min$^{-1}$) |
|---|---|---|---|---|---|---|---|
| 3000 | 5.5 | Continuous | 900–1080 | 60 | 0.6 | 150 | Ar; 15 |

A functional sample of a pressure vessel with a diameter of 270 mm and a length of 1000 mm was welded from the same 5 mm thick steel; see Figure 1b. One longitudinal and two circumferential square butt welds were welded under the same conditions.

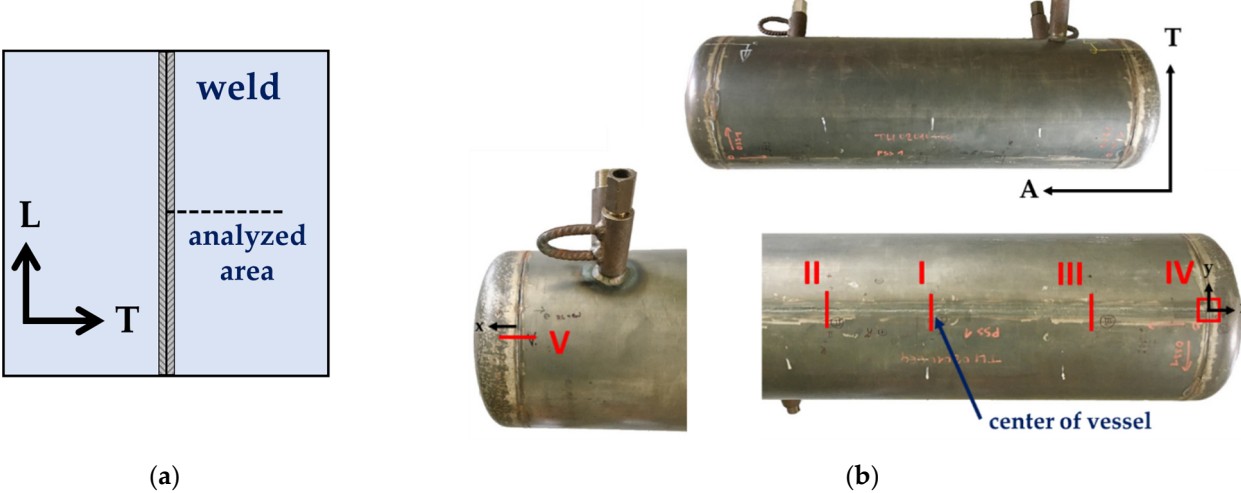

**Figure 1.** (**a**) Scheme of the plate and (**b**) the pressure vessel model with analyzed areas and directions.

## 2.1. Metallography

For metallographic analysis, the plate was cut in the T-N plane. Then, the sample was ground first and then polished. The steel structure was induced by etching in 2% Nital (98 mL ethanol + 2 mL $HNO_3$). The analysis was performed on the Zeiss Axio Observer light microscope (Carl Zeiss Microscopy GmbH, Berlin, Germany). The grain size measurement was carried out according to the ČSN EN ISO 643 standard.

## 2.2. Mechanical Tests

Tensile tests were performed according to the ČSN EN ISO 4136 standard on the EUS 40 testing machine (Werkstoffprüfmaschinen Leipzig GmbH, Leipzig, Germany).

For sample hardness measurements, a Vickers hardness tester and a maximum load of 98.07 N (HV10) were used. According to the ČSN EN ISO 9015-1 standard, there are two different standard methods of evaluation, namely, (1) performing indentations at exact regular distances and (2) a simplified method of performing indentations at each weld zone—the base metal (BM), heat-affected zone (HAZ), weld metal, second HAZ, and BM again. The latter method was used. According to the standards, three indents were made in each zone from which the mean value and standard deviation were calculated.

## 2.3. Microstructure

The microstructure of a material is composed of different phases of variable shape, size, and distribution (grains, precipitates, dendrites, pores, etc.). The X-ray diffraction microstructure parameters include lattice defects, coherently diffracting domain—crystallites, microstrains, and preferred grain orientation—texture [16].

An X'Pert PRO MPD diffractometer (Malvern Panalytical B.V., Almelo, The Netherlands) with cobalt radiation was used for the microstructure parameter estimation by X-ray diffraction (XRD). The crystallite size and microstrain were determined from the XRD patterns using the Rietveld refinement performed in MStruct software (version 2019, Charles University in Prague, Prague, Czech Republic and Lund University, Lund, Sweden) [17]. The crystallite size and microstrain values were also used to calculate the dislocation density using the Williamson and Smallman method [18]. The irradiated volume was defined by experimental geometry of the diffractometer, the effective penetration depth of the X-ray radiation (approx. 5 μm), and the pinhole size (4 mm × 0.25 mm).

Note that X-ray diffraction is not able to directly distinguish ferrite, bainite, and martensite in low-carbon steels because of the small tetragonality of martensite, i.e., the observed diffraction maxima are overlapping. Furthermore, it is necessary to distinguish between grains and crystallites. A crystallite is considered a domain that has an almost

monocrystalline structure with a minimum of defects. Therefore, it is clear that a grain where the spatial orientations of individual parts may differ from each other by several degrees is not the same as a crystallite. Thus, the grain consists of an aggregate of randomly slightly rotated crystallites. Microstrain is caused by the inhomogeneous distribution of elastoplastic deformation on a microscale and is homogeneous in volume within the size of crystallites [19].

### 2.4. Residual Stresses Analyses

Surface macroscopic residual stresses (RS) were described along a line perpendicular to the weld (see Figure 1a) using X-ray diffraction and the Proto iXRD Combo diffractometer (Proto Manufacturing Ltd., Windsor, ON, Canada) with chromium radiation. The values of surface RS were calculated from the lattice deformations, which were determined based on the experimental dependencies of $2\theta(\sin^2\psi)$ assuming a biaxial state of RS, where $\theta$ is the diffraction angle, $\psi$ the angle between the sample surface and the diffracting lattice planes [20]. Diffraction angles $2\theta$ were determined using Gaussian function and Absolute peak method from the peaks of the diffraction lines $K\alpha_1$ of the {211} planes of the $\alpha$-Fe phase. The X-ray elastic constants $\frac{1}{2}s_2$ = 5.76 TPa$^{-1}$, $s_1$ = –1.25 TPa$^{-1}$ were used for the stress calculation using the software XrdWin32 (version 2.0, Proto Manufacturing Ltd., Windsor, ON, Canada). The FWHM parameter denoted the full width at half maximum of $K\alpha_1$ diffraction line for $\psi$ = 0°. FWHM increases with increasing microstrain, dislocation density, and/or decreasing crystallite size [19].

The welded plate was analyzed in the middle area (see Figure 1a). The analyzed areas of the vessel model, see Figure 1b, have been described in the direction perpendicular to the welds, as this direction is more important from a technological point of view. The irradiated volume was defined by experimental geometry of the diffractometer, the effective penetration depth of the X-ray radiation (approximately 4–5 μm), and the size of the collimator (1 mm in diameter with 10 mm oscillation along the weld; except for area IV).

### 2.5. High-Cycle Fatigue Testing

The samples were produced with dimensions corresponding to the ASTM E466 standard. High-cycle fatigue tests were performed on a Schenck PHT resonance machine (Schenck PHT GmbH, Darmstadt, Germany) with a load capacity of 200 kN with Zwick computer-controlled electronics (Zwick Roell Group, Ulm, Germany) and performed at different stress ranges to obtain the whole Wöhler curve, including the endurance limit. The load asymmetry was R = 0.1 and the frequency was 33 Hz. The dog-bone-shaped samples were used for fatigue tests. The weld was located across the center of the length of the analyzed samples where the surface of the sample was welded.

Fatigue and burst pressure tests of the vessel model were performed using special equipment at SVÚM a.s. The pipes and vessels were pressurized with room-temperature water. An essential part of the unique pipeline testing system was the concrete pool in which the pipes and vessels were placed during all tests. The HBM QuantumX MX1615B data acquisition system was used for strain gauge measurements. The data from the pressure sensor, strain gauges, and temperature sensors were measured at a rate of 50 Hz. HBM type 1-LY11-3/120 strain gauges with a 3 mm grid size and self-compensating for steel were placed on the tested pipes. Strain gauges T1–T6 were oriented circumferentially, thus, in the direction of the main stress, since the hoop stress is double the longitudinal stress. Strain gauge T1 was placed in the weld central line. The strain gauge T2 was placed at the fusion boundary of the longitudinal laser weld. Strain gauges T3, T4, and T5 were placed at distances of 10, 20, and 50 mm from the weld, respectively. Strain gauge T6 measured the circumferential stress on the vessel shell.

## 3. Results and Discussions

The standard tests (metallographic, hardness, and tensile) were supplemented with fatigue tests. These findings were compared with non-destructive X-ray diffraction and all

gave an extensive and complex description of not only the weld itself but also the pressure vessel material properties.

### 3.1. Metallography

The macrostructure showed (Figure 2) that width of the weld metal (WM) and the heat-affected zone (HAZ) was approx. 2.2 mm and 4.4 mm, respectively. The microstructure of the base metal (BM) consisted mainly of polygonal ferrite, bainite, and a small fraction of granular perlite (see Figure 3a). The ferrite grains were very fine, with a mean grain size of 8.2 μm (G11 according ČSN EN ISO 643 standard). In addition, elongated sulfides MnS were observed in BM. The closest to the WM was the coarse-grained zone (CGHAZ). This zone was relatively narrow and the microstructure was heated almost to the melting temperature, resulting in the coarsening of the primary austenitic grains and the decomposition of the carbides and carbonitrides of the microalloying elements, which led to the formation of coarse bainite and a small volume fraction of low-carbon martensite. In the inter-critical zone (ICHAZ), see Figure 3b, recrystallized nuclei, heated to temperatures below A3, were primarily formed along the grain boundaries, with occasional nuclei occurring within the grain on impurities. Among CGHAZ and ICHAZ, there was the fine-grained zone (FGHAZ). This zone was the widest compared to other zones in the HAZ and consisted of fine ferrite and bainite (see Figure 3c). The microstructure of the WM was oriented preferentially in the direction of the solidification gradient and consisted of grain boundary ferrite, bainite, and acicular ferrite (see Figure 3d).

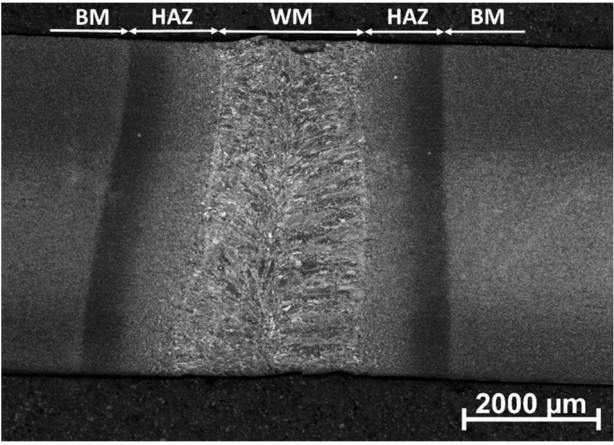

**Figure 2.** Macrostructure of the weld.

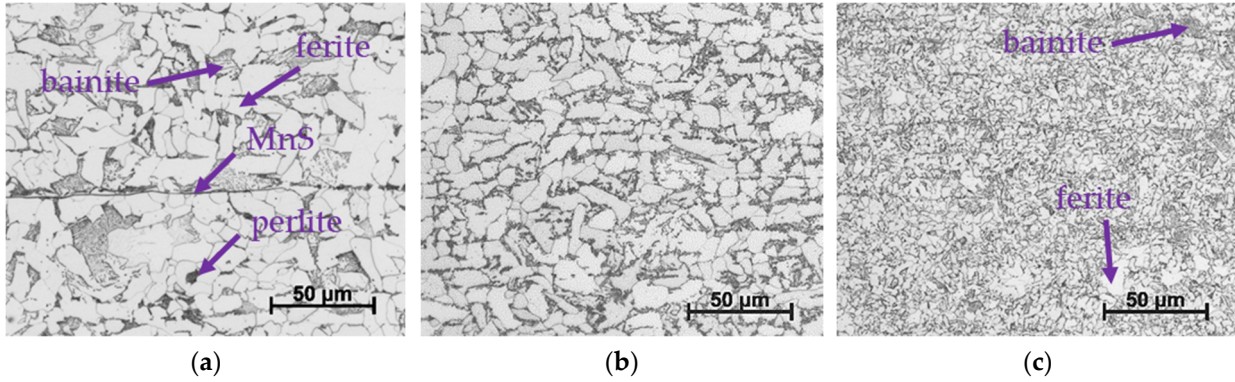

| (a) | (b) | (c) |

**Figure 3.** *Cont.*

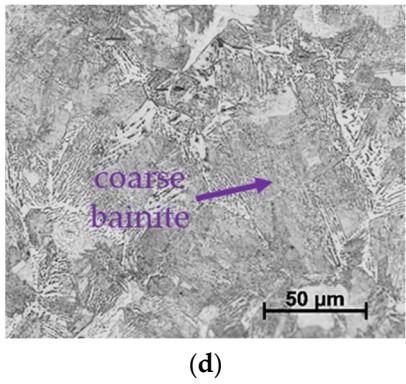

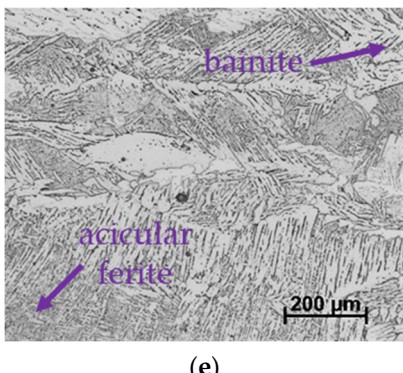

**(d)**　　　　　　　　　　　　　　　　　**(e)**

**Figure 3.** Microstructure of P265GH steel: (**a**) BM, (**b**) ICHAZ, (**c**) FGHAZ, (**d**) CGHAZ, and (**e**) WM.

### 3.2. Mechanical Tests

#### 3.2.1. Tensile Test

From Figure 4a, it can be seen that the first group of samples (S1 and S2) was unsatisfactory because the minimum tensile strength of the BM was not achieved during the tensile test and fracture was observed in the WM, which was the result of imperfect welding in this group of samples. This is an important observation that the parameters of laser welding technology also need constant attention, although it is a technology with, in principle, significantly more stable weld properties.

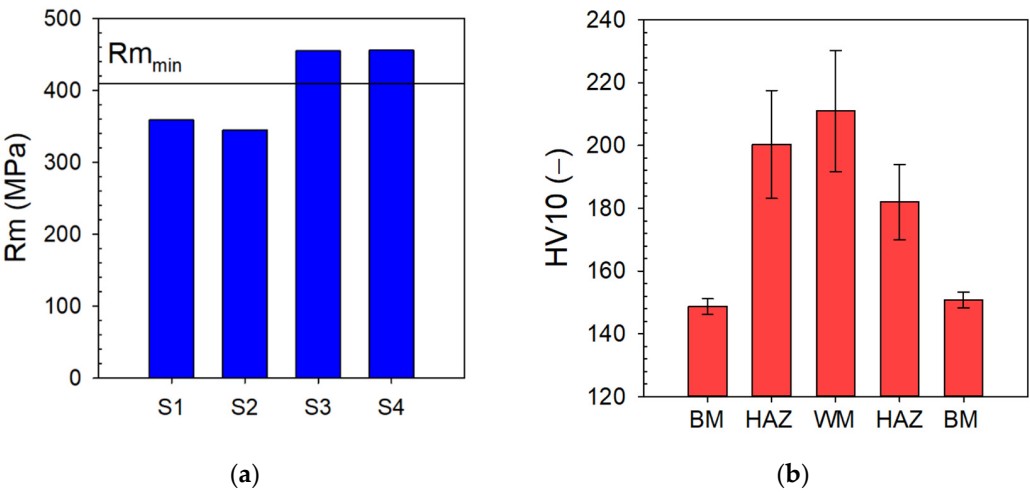

**(a)**　　　　　　　　　　　　　　　　　**(b)**

**Figure 4.** (**a**) Tensile test and (**b**) hardness test results.

After improving the preparation of the samples, such as cleaning the edges and setting the distance between plates, and adjusting the laser welding parameters (routing of shielding gas), a second group of samples was tested and tensile strengths of about 450 MPa were obtained, a result which was about 40 MPa above the minimum tensile strength of the BM and, therefore, a satisfactory result. For samples S3 and S4, the fracture was located in the BM. Therefore, further analyses were performed on the second group of samples.

#### 3.2.2. Hardness

An essential characteristic of welds is the hardness and its course from BM through HAZ and WM. Such a course of hardness HV10 is shown in Figure 4b. Not exceeding the limit value of hardness is one of the fundamental conditions for approving the welding procedure. For the material P265GH, it is specified in the standard ČSN EN ISO 15 614 and TNI CEN ISO 15 608 that the hardness must not exceed 380 HV10.

The diagram also showed a very uniform and smooth course of hardness. In the case of low-alloy steels, microstructural components, such as bainite and martensite, are

usually formed in WM during welding due to high cooling rates. Martensite and bainite are harder compared to ferrite and pearlite, a typical microstructural component of low-alloy steels in BM; therefore, a course of the hardness arises when increases towards WM occur. This course of hardness was also expected in the case of steel P265GH; in addition to the above, it was based on the results obtained on vessel steel P355 during previous works [7,12]. The highest hardness was achieved in WM 211 ± 19 HV10, meeting the mentioned requirements of the standard of not exceeding 380 HV10 due to the different coarse and hardness of phases in the WM. Similarly, a high standard deviation can be observed for HAZ, where there is even more variability in the phases than WM and the measured hardness is quite dependent on the location of the indentation. Luo et al. [21] found by measuring the microhardness HV0.5 that CGHAZ is characterized by relatively high hardness but, on the other hand, ICHAZ is characterized by low hardness.

The lowest hardness was measured for BM 150 ± 2.5 HV10. When the hardness is recalculated to the tensile strength of the material according to the EN standard, good agreement was observed between the hardness measurement and the tensile test performed on the second group of samples (S3 and S4) was observed.

### 3.3. Microstructure Parameters

Resulting from a previous study by Čapek et al. [7], the analysis of the microstructural parameters (i.e., FWHM parameter, crystallite size $D$, microstrain $\varepsilon$, and dislocation density $\rho$) by non-destructive X-ray diffraction in the surface layers indicated that this method could be a good tool for localization of a potential area for surface crack initialization. Therefore, the comparison of microstructural parameters depending on the distance from the weld axis is depicted in Figure 5. In comparison to metallography, see Section 3.1, typical zones were identified:

1. WM (up to approx. 1 mm from the weld axis): The values of $\varepsilon$ and $D$ were the highest, approx. $20 \times 10^{-4}$ and 500 nm, respectively. Therefore, this zone had a coarse-grained microstructure and each grain was distorted due to the quenching of the weld. The diffraction maximum of ferrite and bainite is so close that they cannot be separated from each other. However, their presence, which was confirmed by metallography, causes a higher FWHM value. Therefore, due to the high microstrain values, different phases, and the higher hardness in the WM, the FWHM parameter and the dislocation density (compared with immediate surroundings) had the highest value in the weld axis.

2. HAZ (up to 2–3 mm from the weld axis): A steep decrease in the values of $D$ and $\varepsilon$ values were observed. Parallel with $D$ decreasing, the dislocation density increased resulting from finer-grained microstructure. As was mentioned in the Section 3.1., the CGHAZ was narrow and the closest to the WM. Next to the CGHAZ is FGHAZ, followed by ICHAZ. These zones were not observed in Figure 5, because the resolution of the surface measurements was approx. 0.5 mm, resulting from the divergence of the X-ray beam. According to the metallography, the width of the HAZ from the fusion zone (FZ) was only 0.8 mm.

3. BM with higher $\rho$ (up to approx. 5 mm from the weld axis): The values of $D$ reached the minimum value and $\varepsilon$ with dislocation density $\rho$ gently increasing. In this region, there was a fine-grained microstructure but still with some grain distortion in comparison to BM far from the weld. The temperature was not high enough to enlarge the grains; on the other hand, the influence of heat and the cooling rate were sufficient to retain sufficient energy for grain distortion, but not for the change in the microstructure. This zone overlapped with the so-called stress-affected zone.

4. BM (above 5 mm from the weld axis): All microstructure parameters reached constant values. The material was no longer influenced by welding.

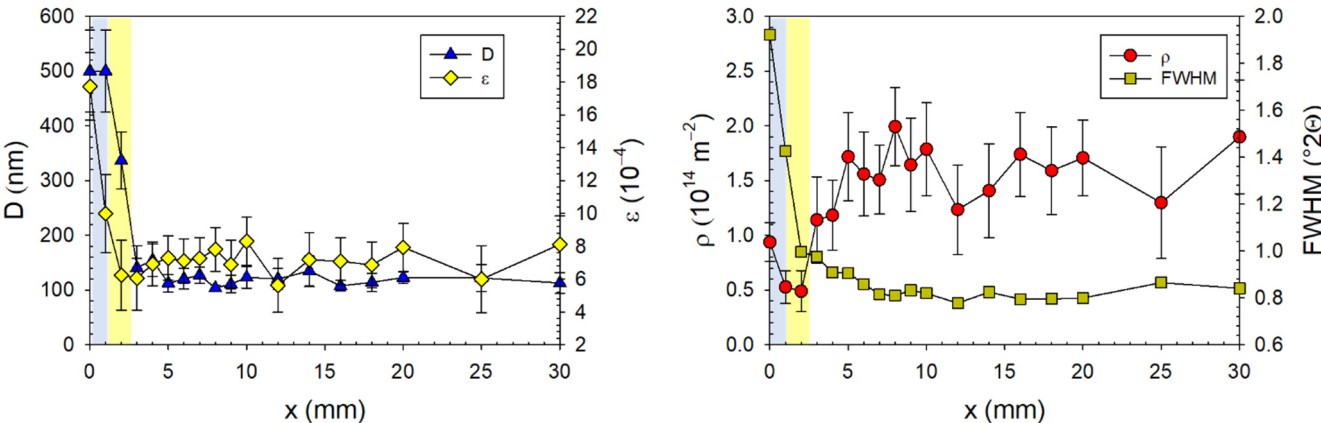

**Figure 5.** Dependence of FWHM parameter, crystallite size *D*, microstrain $\varepsilon$, and dislocation density $\rho$ on the distance from the weld axis. The blue and yellow areas denote the position of the WM and HAZ, respectively.

Two surface microstructural notches, i.e., the steepest microstructure change, were found by XRD in both ranges of HAZ. However, the most important microstructural notch was among the WM and HAZ (i.e., FZ), where all microstructure parameters changed significantly. These areas were the most critical for the potential initialization of surface fatigue crack.

### 3.4. Residual Stresses

In Figure 6, the typical trends of RS occurred in both directions. The RS values in the longitudinal direction had a tensile character as a result of the thermal shrinkage of the WM. In the transversal direction, both compressive and tensile RS were observed; however, due to the transformation effect, the undesirable tensile RS did not exceed 100 MPa.

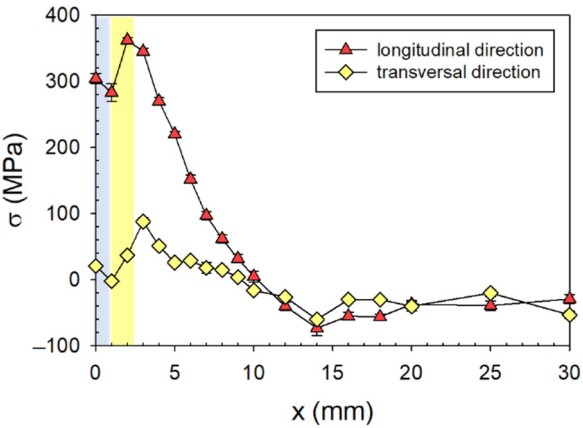

**Figure 6.** Residual stresses depending on the distance from the weld axis on the top side of the plate. The blue and yellow areas denote the position of the WM and HAZ, respectively.

Resulting from the metallography, the width of the WM and the HAZ were approx. 2.2 mm and 3.8 mm, respectively. Nevertheless, from Figure 6, the position of the WM, HAZ, and BM can be distinguished too. According to Čapek et al. [7] and Figure 5, the boundary among the WM and HAZ, the so-called fusion zone, is the most critical area for a potential surface crack initialization because of the surface and microstructural notches. The compressive or insignificant RS were found in the transversal direction in this area and, thus, decelerated the surface crack nucleation, formation, propagation, and, for this reason, prolonged fatigue life.

The RS trends also showed the size of the so-called stress-affected (tempering) zone. This zone did not differ from the BM by microstructure but only by RS and extended approx. 13 mm from the weld axis.

The RS in and near the weld in the L direction exceeded the yield strength (approx. 265 MPa for the BM) and approached the tensile strength (approx. 410 MPa). This effect was caused by the higher hardness of WM and HAZ; see Figure 4b. This indicates the occurrence of phases with higher yield strength. Further, fracture during tensile tests was located in the BM, i.e., WM and HAZ have higher tensile strength. The issue was discussed in more detail in [8].

*3.5. High-Cycle Fatigue Test*

The results of the high-cycle fatigue (HCF) in a semi-logarithmic Wöhler curve are shown in Figure 7. The laser welding parameters were not completely optimized, resulting in insufficient fusion of some parts of the welds. Basically, this was a significant shortcoming, which was, however, eventually turned to positive and useful results, as such a case could occur even in reality. However, note that, according to the certification standards ISO 13 919-1, these weld defects were still within acceptable limits. The main reason for the investigation was to gain the knowledge and data necessary for the evaluation of the safety and reliability of the model pressure vessel described in the following part of this paper, where the results of the fatigue pressure test and residual stresses will be described and discussed.

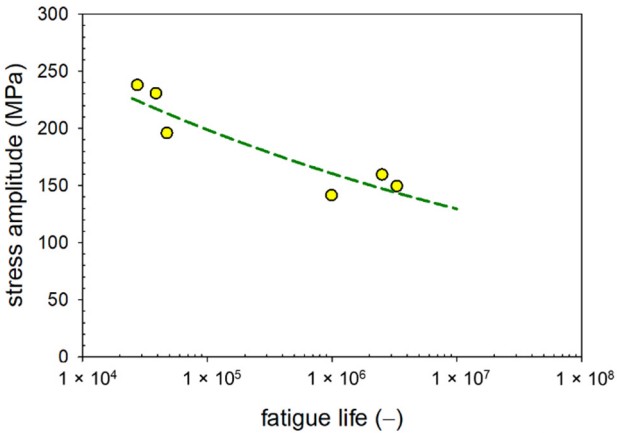

**Figure 7.** Wöhler curve of laser-welded P265GH steel.

Although the samples exhibited some imperfections due to the lack of optimized welding conditions, the scatter was acceptable and comparable to conventional fatigue tests performed on homogeneous materials. In the high-stress amplitudes, slightly below the yield strength of the material, quite satisfactory lifetimes were obtained in the interval from $1 \times 10^4$ to $1 \times 10^5$ cycles. It could be estimated that the fatigue strength was around 130 MPa. For example, if the sample is the model pressure vessel with a diameter of 270 mm and a maximum pressure of 16 bar, then the maximum stress amplitude on the weld is 43 MPa, still safely well below the observed fatigue limit. Note that the fatigue limit corresponded to 29% of the tensile strength, which is lower than the P355NL1 steel analyzed in our previous work [7]. For P355NL1 steel, fatigue strength of 200 MPa was achieved, corresponding to 40% of the tensile strength. The initiation of fatigue cracks occurred in the P265GH steel in WM, particularly the multiple initiation from pores, which is a significant difference compared to P355NL1, where the initiation occurred in the HAZ. As commented in the previous chapters, from the point of view of both RS and microstructure parameters, the FZ is the most critical location in flawless welds; however, in the case of P265GH steel, the effect of defects in WM prevailed. By improving the welding conditions, it would be possible to reach up to 40% of the tensile strength. Slightly lower fatigue strength values

can also be attributed to tensile stresses, see Figure 6, which were described in the loading direction, i.e., the direction perpendicular to the weld.

### 3.6. Vessel Model

#### 3.6.1. Residual Stresses

The final part of this research was the analysis of a functional model part, in this case, the pressure vessel. According to the expected maximum internal pressure during the pressure fatigue test of the model vessel (16 bar), 5 mm thick P265GH steel was selected. The pressure vessel consists of two types of welds—longitudinal and circumferential; see Figure 1b. For both types of welds, the direction perpendicular to the weld is more important from the fatigue point of view. The trends of the surface RS distribution should be similar to those of the RS on the plates, with the fact that it is necessary to take into account the curvature of the plate forming the shell of the vessel and the different production technology of the heads by forging.

The values of the RS in the longitudinal weld, see Figure 8a, showed favorable courses, since high tensile RS values were not observed, which would be unfavorable from the fatigue life point of view. The detailed course of the RS of the longitudinal weld, see Figure 8b, showed the high compressive RS in the HAZ and the BM. This was probably due to the uneven bending of the plate before welding. The maximum value of FWHM at 10 mm was given by a combination of the change in the microstructure of the material among the HAZ and the BM and the grinding of the edges prior to welding.

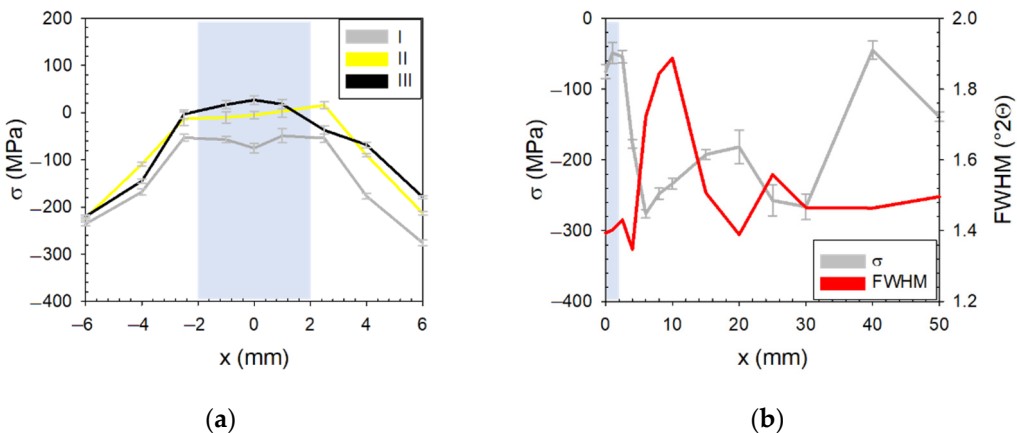

**Figure 8.** (**a**) Macroscopic residual stresses of the longitudinal weld in the transversal direction and (**b**) macroscopic residual stresses and FWHM parameter of the area I. The blue area denotes the position of the longitudinal weld [22].

Figure 9 describes the RS and FWHM parameter of the circumferential weld, when the positive values of the X-axis are the head side. The analysis direction was always, also for the crossing area, perpendicular to the circumferential weld. From the point of view of the service life, the RS of the circumferential welds are favorable and reached only low tensile values, approx. 100 MPa. Values of the FWHM parameter were similar for both heads; however, the crossing area showed lower values for negative x (WM of the longitudinal weld), mainly due to the different microstructures because of the thermal influence of the crossing welds.

Figure 10 shows the results of RS and FWHM parameter for the crossing of the circumferential and longitudinal welds, where the analyses were performed perpendicular to the circumferential weld. The WM itself (bordered with dashed lines) showed favorable compressive RS, except for the area of the crossing and the upper border of the longitudinal weld. The upper border of the longitudinal weld is also critical from the point of view of microstructural parameters, since in the same place, there are high values of the FWHM parameter, which means high microstrain, dislocation density, and/or small crystallite size.

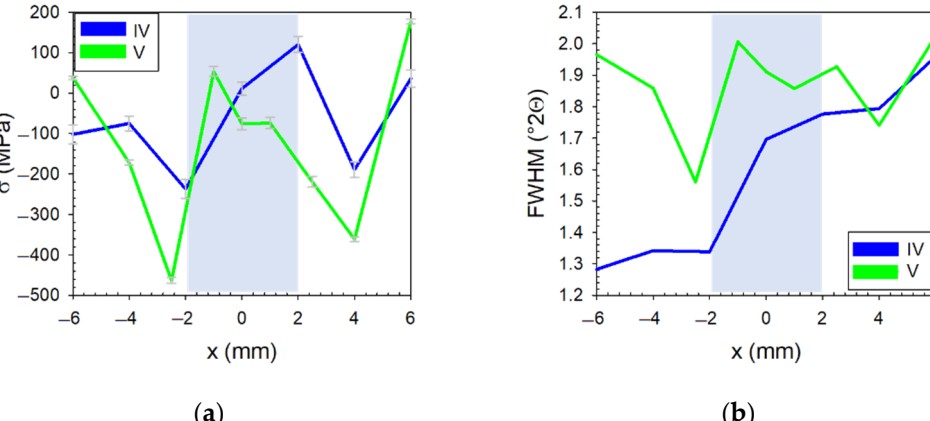

**(a)**                                                  **(b)**

**Figure 9.** (**a**) Macroscopic residual stresses and (**b**) FWHM parameter of the circumferential weld in the axial direction. The blue area denotes the position of the circumferential weld [22].

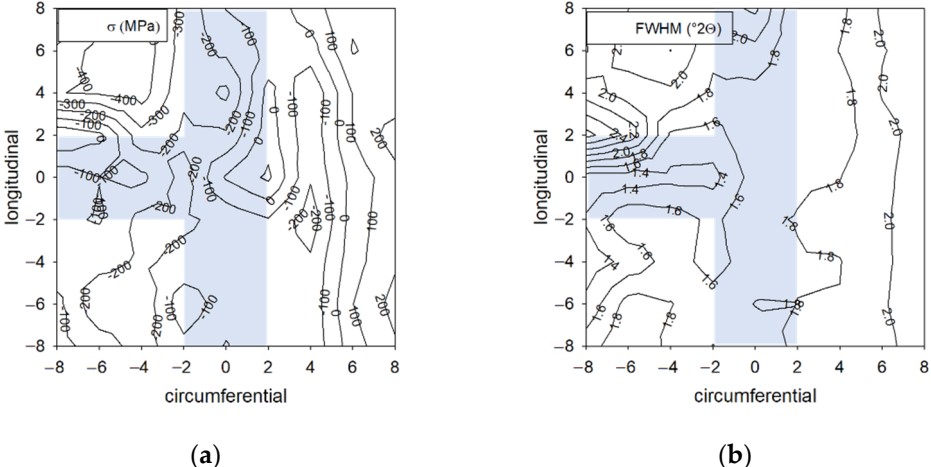

**(a)**                                                  **(b)**

**Figure 10.** Maps of the (**a**) macroscopic residual stresses and (**b**) FWHM parameter of the crossing area of the longitudinal and circumferential welds of the pressure vessel (blue area). The analysis direction is perpendicular to the circumferential weld [22].

The surface RS of the BM in the bent plate of the vessel shell were also compressive, but we observed different values of both the RS and the FWHM parameter on both sides of the shell of the weld. This is probably due to uneven bending of the plate. On the contrary, the tensile RS and slightly higher values of the FWHM parameter were found for the BM of the head. The different manufacturing of the hot-rolled plate and the forged head mainly causes this difference. However, the tensile surface macroscopic RS of the head do not affect the fatigue resistance because the fatigue cracks are usually initialized at the boundary or in the weld itself.

### 3.6.2. High-Cycle Fatigue Test

The high-cycle fatigue pressure test of the model vessel was carried out at a pressure from 4 to 16 bar, with a total number of 1 million cycles. Pressure was chosen taking into account the operating parameters of industrial natural-gas- and coal-fired hot-water boilers, which operate under similar pressure loads. A record of three pressure cycles is shown in Figure 11a. As can be seen from the curve, one cycle took approximately 1.5 s, which corresponds to a frequency of 0.65 Hz. The stress–strain curves are shown in Figure 11b, where there is visible hysteresis, but without any propagation during the high-cycle fatigue test. The maximum measured strain was converted to stress using Hooke's law and it is important to note that conversion to stress using Hooke's law is only valid in the elastic region. Young's modulus of elasticity for steel 210 GPa was used for the conversion. The

dependence of applied stress on the distance from the weld is shown in Figure 12. The highest applied stress could be observed in the WM around 130 MPa, which corresponded to the determined fatigue strength. At the FZ, a significant stress drop was visible, down to a value of 40 MPa. At a distance of 10 mm from the weld, an increase in stress to a level just above 80 MPa can be seen again. Far from the WM, a gradual drop in applied stress can be seen. If the applied stress and residual stresses (from Figure 8b) are added up, the maximum of total stress is in the weld itself, when the stresses are slightly tensile, which is unfavorable from the fatigue life point of view. Therefore, the area between the weld axis and 10 mm from the weld showed stress peaks; the relation to the variability in RS and FWHM, see Figure 8, in the HAZ can be considered a critical place for the formation of fatigue and thermomechanical cracks during the future operation of hot water boilers. Thus, it is necessary to pay close attention to this area during the shutdown of boiler operation and to carry out a non-destructive control for the presence of crack-like defects.

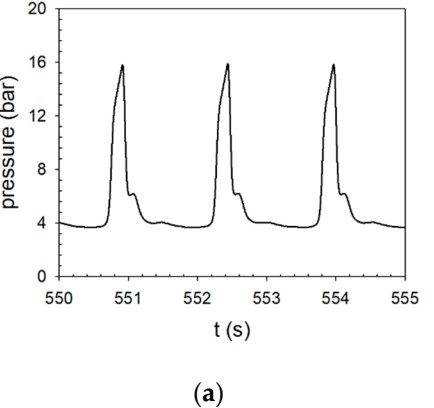

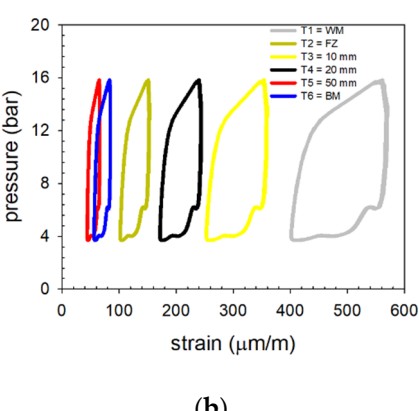

(**a**)                                   (**b**)

**Figure 11.** (**a**) Pressure-time and (**b**) pressure-strain curve at high-cycle fatigue pressure test.

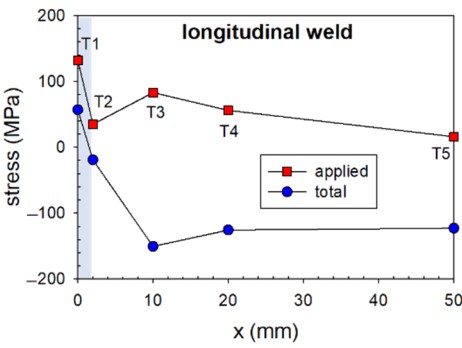

**Figure 12.** Applied stress distribution and the total stress (sum of the applied and residual stresses) at pressure 16 bar as a function of the distance from the weld axis. The blue area denotes the position of the corresponding weld.

After passing the high-cycle fatigue pressure test, a burst pressure test was performed to verify the residual strength of the vessel. The burst pressure test was carried out by stepwise loading until destruction. In Figure 13a, the pressure–time curve can be seen; the destruction of the vessel occurred at a pressure of 134 bar, according to Barlow's formula, corresponding to a hoop stress of 360 MPa. The first significant yield can be observed in the vessel shell (strain gauges T5 and T6) above pressure of 120 bar; see Figure 13b. Due to the first observed yield of the vessel, a longer pause was made to allow full strain flow. Above 120 bar, the strain gauges T5 and T6 showed a notable strain gain on the vessel shell, reaching almost 8000 μm/m. Although the strain gauges T1 and T2 showed mostly elastic deformation, apparently, the fact that the conventional yield strength corresponds to 0.2% of the plastic deformation, i.e., 2000 μm/m, was taken into account. It can be stated that the residual strength of the model vessel was high, even after the severe fatigue

test. The destructive pressure corresponds to 82% of the tensile strength obtained in the standardized samples made from welded plates, which can be considered as a significant result. Although it was analyzed as a weak point of the fusion zone on the plate, the fracture occurred in the weld of the pressure vessel itself. This was most likely caused by macroscopic notches, i.e., defects, and an unfavorable distribution of total stresses, see Figure 12. The results prove that laser-welded vessels show high resistance to static and fatigue loads.

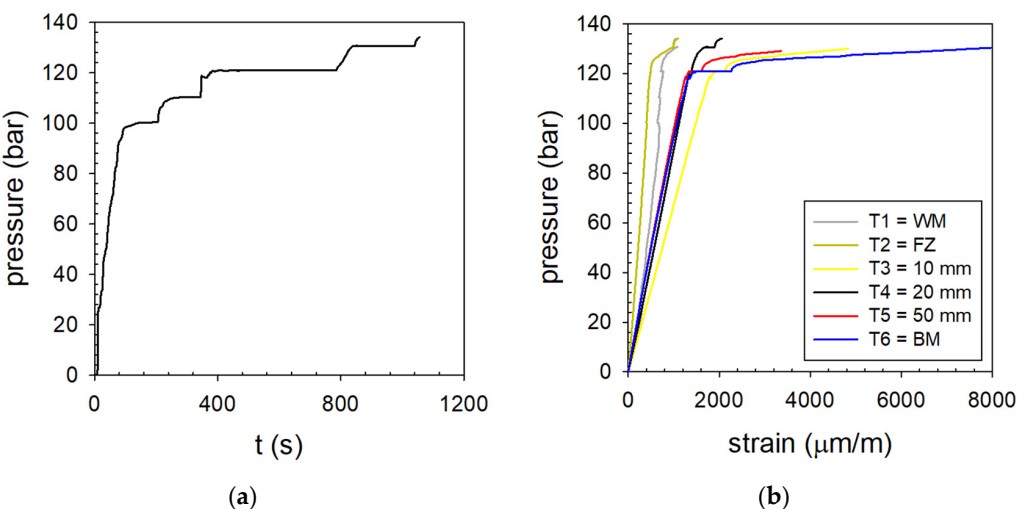

**Figure 13.** (**a**) Pressure–time and (**b**) pressure–strain curve at burst test.

## 4. Conclusions

The purpose of this investigation was to explain the relationship of the microstructure, mechanical properties, and residual stresses of laser-welded heat-resistant pressure vessel steel plates. Specifically, 5 mm thick square butt welds were investigated on P265GH steel plates for pressure vessels and boilers. Various experimental techniques were used for the analysis of the weld properties and critical areas for a potential surface crack initialization. Furthermore, the functional model of the pressure vessel subsequently verified the results and assumptions. The major experimentally obtained knowledge can be summarized in the following points:

- Through metallographic analysis and XRD, three fundamental areas of laser welds were distinguished: WM, HAZ, and BM. On the boundaries, where the investigated parameters change significantly, there are microstructural notches, critical areas for the potential surface crack initialization. In the flawless weld, the main microstructural notch was determined in the FZ.
- Tensile testing and fatigue resistance on standardized samples contributed to the improvement in laser welding parameters. For samples without significant weld imperfections, the fatigue strength was estimated at 130 MPa, which corresponded to 29% of the tensile strength. Both the tensile and fatigue samples with major weld imperfections failed in WM.
- Based on these results, the model pressure vessel with a diameter of 270 mm and a maximum pressure of 16 bar theoretically reaches the maximum applied stress of 43 MPa (and, in this case, total stress too) in the weld, still safely well below the observed fatigue limit.
- The experience gained from the tests and analyses was applied to the welding of a model pressure vessel, which was fatigued and statically loaded in the conditions corresponding to operation. Based on the strain measurements, it was found that the highest stress values occurred in the central line of the we ld around 130 MPa. From the Wöhler curve of the welded plate, it was found that this is the fatigue strength. However, after including the RS, the total stresses in the WM and FZ were reduced;

furthermore, no propagation was observed on the strain gauges during fatigue testing and 1 million cycles were achieved.

- Subsequent static testing to destruction revealed excellent static strength after a severe fatigue test. The failure occurred at 134 bar, which corresponded to 82% of the tensile strength obtained in the standardized samples made of welded plates.

The results prove that laser-welded vessels show high resistance to static and fatigue loads. Based on the results of this research, it would be possible to continue the development of laser welding technology for pressure vessels and boilers, which would find applications in production.

**Author Contributions:** Conceptualization, J.Č., K.T., J.K. and I.Č.; resources, I.Č., N.G. and S.N.; methodology, investigation, and writing—original draft preparation, J.Č., K.T. and J.K.; writing—review, editing, and visualization, J.Č., K.T., J.K., I.Č., N.G., K.K. and S.N.; supervision, J.Č. and I.Č.; project administration, data curation and funding acquisition, I.Č. All authors have read and agreed to the published version of the manuscript.

**Funding:** This work was supported by the Center for Advanced Applied Science, grant number CZ.02.1.01/0.0/0.0/16_019/0000778 "Center for Advanced Applied Science" within the Operational Program Research, Development and Education supervised by the Ministry of Education, Youth and Sports of the Czech Republic and the project TH02010664 of the Technology Agency of the Czech Republic. K.T.'s work was supported by the Grant Agency of the Czech Technical University in Prague, grant number SGS22/183/OHK4/3T/14.

**Institutional Review Board Statement:** Not applicable.

**Informed Consent Statement:** Not applicable.

**Data Availability Statement:** Data sharing is not applicable to this article.

**Conflicts of Interest:** The authors declare no conflict of interest.

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
