# Peer review of "Fatigue Properties and Residual Stresses of Laser-Welded Heat-Resistant Pressure Vessel Steel, Verification on Vessel Model"

_metals, doi:10.3390/met12091517_

Round 1
Reviewer 1 Report
The heat-resistant pressure vessel steel was carried out by using laser welding, and the microstructure, the fatigue properties and residual stresses were studied. Moreover, the model was employed in the research. The word is interesting, however, some parts need to be verified.
1. In abstract, the background should be declined to one sentence with at most two line of words. Then, to introduce your research purpose, method and result. The main content in the abstract is to show your results and data including the evaluation of the microstructure, the change of the fatigue properties and the residual stresses, and so on.
2. Please write the full name for the first abbreviation, such as WM, HAZ in line 59, XRD in line 62, et. al.
3. Please show the detail welding parameters, such as laser spot size, defocus length, the angle of inclination, the shielding gas and its flow.
4. It is better to show the overall morphology of the weld joint in Fig. 2 in order to prove the data in line 215 and 216. Moreover, the ferrite, bainite and perlite should be marked in Fig. 2.
5. In 241, please explain how to adjust laser welding parameter and indicate the difference between this parameter and Table 2. Fig. 3(a) should give the key condition in S1, S2, S3 and S4. In addition, please give the width of the areas in Fig. 3(b) and explain the meaning of ordinate unit.
6. It is necessary to show the governing equation or boundary condition in the model. And explain how to verify the reliability of this model.
7. Please decrease the content of the conclusions which only shows the highlights of this study.
Author Response
Dear Madam or Sir,
I appreciate your review.
Please see the attachment.
Yours faithfully
Jiri Capek

Reviewer 2 Report
This paper adresses the scientific question whether the manufacture of power plants can be made more cost-effective by using laser-welded components instead of seamless pipelines. To accomplish this, it must be ensured that laser-welded heat-resistand pressure vessel steel seams, meet the high demands on mechanical properties and durability for the usage in boilers, pressure vessels and pipelines. Therefore, the researchers perform metallographic analysis, mechanical tests, microstructure investigations, residual stress analysis as well as fatigue testing. Furthermore, they ensure the transferability of the results to real-life applications by using a Vessel Model. Overall the presented results are promising.
In general, the paper presents the research topic holistically. Some parts could be more detailed to better highlight the relevance and novelty of the research. The methodological approach described is purposeful. In the following, individual points of criticism are dealt with in more detail
Introduction
The beginning of the introduction is well structured. Unlike the abstract, in line 47 the disadvantages of seamless components and how laser-welded components could improve these issues are missing. The paper would benefit and gain interest if the purpose of the presented research is more clearly explained.
In line 66, a source which supports the statement that laser beam welding is superior over arc welding is missing.
The synthesis of the literature is not as developed as I would expect to see in an academic journal article. Although literature on the topic has been summarized, a derivation of a research gab, which is adressed in the presented work, is missing (line 82-93).
For a better understanding of the context of the paper it would be beneficial to state the problems of integrity of welded structures in line 94.
Results and Discussion
Metallographie
For a better understanding, figure 2 could be illustrated in a different way. F.e the images could be arranged next to each other in the order of their appearance starting from the welded metal. For complettness an image of the microstructure of CGHAZ would be good.
Tensile test
In line 241 it would be interessting how the preparation and welding parameters where improved. Especially, because it is stated in the introduction that process parameters have a strong influence on weld quality. Furthermore, it must be stated, if the in table 2 given welding parameters are the ones used for S1 and S2 or for S3 and S4.
Hardness
For understanding reasons it would be helpful if a detailed explanation was given, why the lowest hardness is expected in the BM region.
Microstructure parameters
In this section it should be checked whether all statements about the displayed graphs are formulated correctly. F.e line 279-280: „… the dislocation density had the highest value in the weld axis.“, Figure 4 b shows that the highest values are in the BM.
In line 278 macrostrains instead of microstrains is written.
High-cycle fatigue test
Line 336-337: To reinforce the statement, the mentioned acceptable limits should be specified in more detail and supported with a source. The same applies to line 345 – 346. Especially a source for comparable fatigue tests is missing.
In general, all figures in section 4 could be described and discussed in more detail.
Conclusion
In the conclusion the main findings from section 4 are well summarized. To link to the introduction and to emphasize the added value of the work, a final comment is missing whether the aim of the research was achieved and whether the results support the assumption that laser welded components can be used in energy plants or not.
Author Response

(The authors gave the same response as above.)
